# GRAPH ANALYSIS AND GRAPH POOLING IN THE SPATIAL DOMAIN

## ABSTRACT

The spatial convolution layer which is widely used in the Graph Neural Networks (GNNs) aggregates the feature vector of each node with the feature vectors of its neighboring nodes. The GNN is not aware of the locations of the nodes in the global structure of the graph and when the local structures corresponding to different nodes are similar to each other, the convolution layer maps all those nodes to similar or same feature vectors in the continuous feature space. Therefore, the GNN cannot distinguish two graphs if their difference is not in their local structures. In addition, when the nodes are not labeled/attributed the convolution layers can fail to distinguish even different local structures. In this paper, we propose an effective solution to address this problem of the GNNs. The proposed approach leverages a spatial representation of the graph which makes the neural network aware of the differences between the nodes and also their locations in the graph. The spatial representation which is equivalent to a point-cloud representation of the graph is obtained by a graph embedding method. Using the proposed approach, the local feature extractor of the GNN distinguishes similar local structures in different locations of the graph and the GNN infers the topological structure of the graph from the spatial distribution of the locally extracted feature vectors. Moreover, the spatial representation is utilized to simplify the graph down-sampling problem. A new graph pooling method is proposed and it is shown that the proposed pooling method achieves competitive or better results in comparison with the state-of-the-art methods.

## 1 INTRODUCTION

Many of the modern data are naturally represented by graphs (Bronstein et al., 2017; Cook & Holder, 2006; Dadaneh & Qian, 2016; Qi et al., 2017b) and it is an important research problem to design neural network architectures which can work with graphs. The adjacency matrix of a graph exhibits the local connectivity of the nodes. Thus, it is straightforward to extend the local feature aggregation used in the Convolutional Neural Networks (CNNs) to the Graph Neural Networks (GNNs) (Atwood & Towsley, 2016; Bruna et al., 2014; Fey et al., 2018; Gilmer et al., 2017; Niepert et al., 2016; Simonovsky & Komodakis, 2017; Zhang et al., 2018). However, when the nodes of the graphs are not labeled/attributed or the labels/attributes of the nodes do not carry information about the differences between the nodes or any information about their locations in the graph, the GNN can fail to extract discriminative features. The GNN maps nodes whose corresponding local structures are similar to same/close feature vectors in the continuous feature space. Although mapping nodes with similar local structures to similar feature vectors might be desirable in some applications (Henderson et al., 2012), this feature can make the GNN unable to distinguish graphs whose difference is not in their local structures. In addition, when the nodes are not labeled/attributed, the spatial graph convolution only propagates information about the degree of the nodes which might not lead to extracting informative features about the topological structure of the graph. Another limitation of the current GNN architectures is that they are mostly unable to do the hierarchical feature learning employed in the CNNs (He et al., 2016; Krizhevsky et al., 2012). The main reason is that graphs lack the tensor representation and it is difficult to measure how accurate a subset of nodes represent the topological structure of the given graph.

**Summary of Contributions.** In this paper, we focus on the graph classification problem. The main contributions of this paper can be summarized as follows.

• A shortcoming of the GNNs is discussed and it is shown that the existing GNNs can fail to learn to perform even simple graph analysis tasks. It is shown that the proposed approach which leverages a spatial representation of the graph effectively addresses the shortcoming of the GNN. Several new experiments are presented to demonstrate the shortcoming of the GNNs and they show that providing the geometrical representation of the graph to the neural network substantially improves the capability of the GNN in inferring the structure of the graph.

• The geometrical representation of the graph is leveraged to design a novel graph pooling method. The proposed approach simplifies the graph down sampling problem into a column/row sampling problem. The proposed approach samples a subset of the nodes such that they preserve the structure of the graph. It is shown that the proposed approach achieves competitive or better performance in comparison with the existing methods.

**Notation.** Given a vector $\mathbf{x}$, $\|\mathbf{x}\|_p$ denotes its $\ell_p$ Euclidean norm, and $\mathbf{x}(i)$ denotes its $i^{\text{th}}$ element. Given a matrix $\mathbf{X}$, $\mathbf{x}_i$ denotes the $i^{\text{th}}$ row of $\mathbf{X}$. A graph with $n$ nodes is represented by two matrices $\mathbf{A} \in \mathbb{R}^{n \times n}$ and $\mathbf{F} \in \mathbb{R}^{n \times d_f}$, where $\mathbf{A}$ is the adjacency matrix, $\mathbf{F}$ is the matrix of node labels/attributes, and $d_f$ is the dimension of the attributes/labels of the nodes. The operation $\mathbf{A} \Leftarrow \mathbf{B}$ means that the content of $\mathbf{A}$ is set equal to the content of $\mathbf{B}$. If $\mathcal{I}$ is a set of indices, $\mathbf{X}_{\mathcal{I}}$ is the matrix of the rows of $\mathbf{X}$ whose indexes are in $\mathcal{I}$. Vector $\mathbf{y} = \text{softmax}(\mathbf{x})$ is defined as $\mathbf{y}(i) = \frac{\exp(\mathbf{x}(i))}{\sum_k \exp(\mathbf{x}(k))}$. The local structure corresponding to a node is the structure of the graph in the close neighbourhood of the node.

## 2 RELATED WORK

This paper mainly focuses on the graph classification problem. In the proposed approach, the geometrical representation of the graph provided by a graph embedding algorithm is utilized to make the neural network aware of the topological structure of the graph. In this section, some of the related research works in GNN and graph embedding are briefly reviewed.

*Graph Embedding:* A graph embedding method aims at finding a continuous embedding vector for each node of the graph such that the topological structure of the graph is encoded in the spatial distribution of the embedding vectors. The nodes which are close on the graph or they share similar structural role are mapped to nearby points in the embedding space and vice versa. In general, the graph embedding methods fall into three broad categories: matrix factorization based methods (Ahmed et al., 2013; Belkin & Niyogi, 2002; Cao et al., 2015; Ou et al., 2016; Roweis & Saul, 2000), random-walk based approaches (Grover & Leskovec, 2016; Perozzi et al., 2014), and deep learning based methods (Wang et al., 2016). In this paper, we use the DeepWalk graph embedding method (Perozzi et al., 2014) which is a random walked based method.

*Graph Neural Networks:* In recent years, there has been a surge of interest in developing deep network architectures which can work with graphs (Bronstein et al., 2017; Bruna et al., 2014; Duvenaud et al., 2015; Fout et al., 2017; Gilmer et al., 2017; Hamilton et al., 2017; Kipf et al., 2018; Kipf & Welling, 2017; Li et al., 2016; Niepert et al., 2016; Simonovsky & Komodakis, 2017; Tixier et al., 2019). Local connectivity, weight sharing, and shift invariance of the convolution layer in the CNNs have led to remarkable achievements in computer vision and natural language processing (Goodfellow et al., 2016). Accordingly, there are remarkable works focusing on the design of graph convolution layers encoding the traditional properties. Most of the existing graph convolution layers can be loosely divided into two main subsets: the spatial convolution layers and the spectral convolution layers. The spectral methods are based on the generalization of spectral filtering in graph signal processing (Bruna et al., 2014; Defferrard et al., 2016; Henaff et al., 2015; Levie et al., 2017). The major drawback of the spectral based methods is the non-generalizability to data residing over multiple graphs which is due to the dependency on the basis of the graph Laplacian. In contrast to spectral graph convolution, the spatial graph convolution performs the convolution directly in the nodal domain and can be generalized across graphs (Nguyen et al., 2018; Niepert et al., 2016; Schlichtkrull et al., 2018; Simonovsky & Komodakis, 2017; Veličković et al., 2017; Verma & Zhang, 2018; Zhang et al., 2018). If the sum function is used to aggregate the local feature vectors, a simple spatial convolution layer can be written as

$$\mathbf{X} \Leftarrow \mathbf{A} \, f\left(\mathbf{X}\mathbf{\Phi_1}\right) + f\left(\mathbf{X}\mathbf{\Phi_2}\right) \quad , \tag{1}$$

where $f(\cdot)$ is the element-wise non-linear function. Some papers use a normalized version of (1) via multiplying (1) with the inverse of the degree matrix (Ying et al., 2018). The weight matrix

$\mathbf{\Phi_2}$ transforms the feature vector of the given node and $\mathbf{\Phi_1}$ transforms the feature vectors of the neighbouring nodes. A drawback of the spatial convolution is that the aggregation function might not be an injective function, i.e., two nodes with different local structures can be mapped to the same feature vector by the convolution layer. The authors of (Xu et al., 2019) showed that with a minor modification, the sum aggregation function can become an injective function. However, when the nodes are not labeled/attributed or the labels/attributes do not inform the GNN about the location of the nodes, the GNN is not able to distinguish graphs whose local structures are similar but their global structures are different. The approach proposed in our paper not only addresses the problem raised in (Xu et al., 2019), it also helps the GNN to distinguish the similar local structures in different locations of the graph.

In order to obtain a global representation of the graph, the local feature vectors obtained by the convolution layers should be aggregated into a final feature vector. The element-wise max/mean functions are used widely to aggregate all the local feature vectors. Inspired by the hierarchical feature extraction in the CNNs, tools have been developed to perform nonlocal feature aggregation (Duvenaud et al., 2015; Luzhnica et al., 2019). In (Zhang et al., 2018), the nodes are ranked and the ranking is used to build a sequence of nodes using a subset of the nodes. Subsequently, a 1-dimensional CNN is applied to the sequence of the nodes to perform non-local feature aggregation. However, the way that (Zhang et al., 2018) builds the sequence of the nodes is not data driven. In addition, a 1-dimensional array might not have the capacity to preserve the topological structure of the graph. In (Ying et al., 2018), a soft graph pooling method was proposed which learns a set of cluster centers that are used to down-sample the extracted local feature vectors. The graph pooling methods proposed in (Gao & Ji, 2019; Lee et al., 2019) are similar to the soft graph pooling method presented in (Ying et al., 2018) but they learn one cluster center which is used to rank the nodes and they sample the nodes which are closer to the learned cluster center. In (Defferrard et al., 2016; Fey et al., 2018), the graph clustering algorithms were used to perform graph down-sizing.

## 3 THE SHORTCOMING OF THE GNNS

Suppose the nodes of the given graphs are not labeled/attributed or assume that the labels/attributes do not contain any information about the role/location of the nodes in the global structure of the graph. Therefore, if two different nodes are not labeled or they are labeled similarly, they appear the same to the GNN. Accordingly, if the local structures corresponding to two nodes are similar, the convolution layer of the GNN maps them to the same feature vector in the continuous feature space. This means that if the local structures of two different graphs are similar, the GNN cannot extract discriminative features to distinguish them. In the followings, we study an example to clarify this feature of the GNN.

*Illustrative example:* Suppose we have a dataset of clustered unlabeled graphs (nodes and edges are not labeled/attributed) and assume that the clusters are not topologically different (a common generator created all the clusters). Every node is connected to small a set of the other nodes in its corresponding cluster. One class of the graphs consist of two clusters and the other class of graphs consist of three clusters. Therefore, the task is to infer if a graph is made of two clusters or three clusters. Consider the simplest case in which there is no connection between the clusters. Assume that we use a typical GNN which is composed of multiple spatial convolution layers and a global element-wise max/mean pooling layer. Suppose that a given graph belongs to the first class, i.e., it consists of two clusters. Define $\mathbf{v}_1$ and $\mathbf{v}_2$ as the aggregation of the local feature vectors corresponding to the first and the second clusters, respectively. The global feature vector of this graph is equal to the element-wise mean/max of $\mathbf{v}_1$ and $\mathbf{v}_2$. Clearly, $\mathbf{v}_2$ can be indistinguishable from $\mathbf{v}_1$ since the clusters are generated using the same generator and the GNN is not aware of the location of the nodes. Therefore, the feature vector of the whole graph can also be indistinguishable from $\mathbf{v}_1$. The same argument is also true for a graph with three clusters. Accordingly, the representation obtained by the GNN is unable to distinguish this two classes of graphs. If one trains a GNN on this task, the test accuracy is around 50 % which is not better than random guess. The main reason is that the local structures corresponding to all the nodes in a graph are similar. In addition, the local structures in a graph with two clusters is not different from the local structures in a graph with three clusters. Since the GNN is not aware of the location of the nodes, all the nodes are mapped to similar

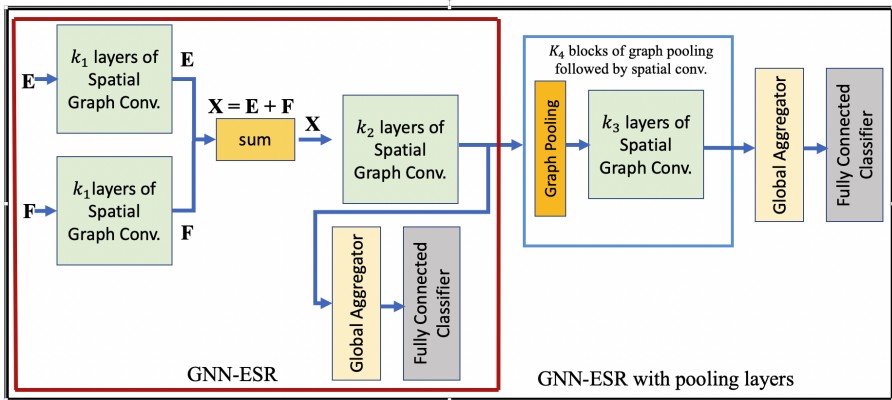

Figure 1: The red box demonstrates a GNN Equipped with Spatial Representation (GNN-ESR). Matrix $\mathbf{E}$ is the matrix of node embedding vectors and $\mathbf{F}$ is the matrix of node labels/attributes. The large black box depicts a GNN in which the proposed graph pooling layer is utilized. In the presented numerical experiments, we set $k_1 = 1$, $k_2 = 2$, $k_3 = 2$, and $k_4 = 1$, i.e., the GNN-ESR is composed of three spatial convolution layers and the GNN-ESR with graph pooling includes one graph down-sampling layer and two convolution layers are placed next to the pooling layer.

feature vectors in the feature space. Accordingly, the spatial distribution of the local feature vectors of the two classes of the graphs are indistinguishable. [1]

Table 1: Classification accuracy while nodes and edges are not labeled/attributed. Both the GNN and the GNN-ESR are composed of three spatial convolution layers and a global max pooling layer.

|  | PROTEINS | NCI1 | DD | ENZYM | SYNTHIE | HLLD | CNLC | CNC |
|---|---|---|---|---|---|---|---|---|
| GNN | 76.87 | 69.09 | 75.68 | 43.30 | 67.75 | 54.24 | 37.33 | 36.40 |
|  | $\pm 4.95$ | $\pm 3.47$ | $\pm 3.45$ | $\pm 5.27$ | $\pm 2.44$ | $\pm 4.15$ | $\pm 3.78$ | $\pm 4.41$ |
| GNN-ESR | 79.47 | 73.72 | 79.77 | 53.51 | 71.15 | 99.10 | 98.18 | 99 |
|  | $\pm 4.11$ | $\pm 2.36$ | $\pm 2.65$ | $\pm 3.63$ | $\pm 2.54$ | $\pm 0.88$ | $\pm 1.05$ | $\pm 0.71$ |
| Improvement | 3.38 % | 6.7 % | 5.40 % | 23.5 % | 5.01 % | 79.4 % | 156.1 % | 179.6 % |

## 4 PROPOSED APPROACH

In section 3, it was shown that the GNN can fail to infer the topological structure of the graph when the local structures in the graph are similar. The GNN is not aware of the location of each node in the global structure of the graph and it maps all the nodes whose corresponding local structures are similar to same/close points in the feature space. This was the main reason that the GNN failed to learn to perform the task described in Section 3. *Ideally, if the extracted feature vector for each node is a function of its location in the graph, the nodes are mapped to different points (corresponding to their locations) in the feature space and the GNN can distinguish the graphs via analyzing the spatial distribution of the extracted local feature vectors.* Accordingly, we propose an approach using which the extracted feature vectors depend on the location/role of the nodes in the global structure of graph. Another motivation for the proposed approach is the remarkable success of the neural networks in analysing point-clod data (Qi et al., 2017a). In point-cloud data, each data point is corresponding to a point on the surface of the object and the location of each point in the 3D space is included in the feature vector which represents each point of the surface. The neural networks which process the point-cloud representation yielded the state-of-the-art performance in the 3D vision tasks (Qi et al., 2017b).

---

[1] As another example, note the last three columns of Table 1 which are corresponding to three classification tasks described in Section 5. The second row shows the accuracy of a GNN on these tasks. In all the three tasks, the performance of the GNN is comparable to random guess.

In order to include the locations of the nodes in the extracted local feature vectors, first we have to define a representation for the location of each node. Suppose that vector $\mathbf{e}_i \in \mathbb{R}^{d_e}$ in the $d_e$-dimensional Euclidean space contains the information about the location of the $i^{\text{th}}$ node. Evidently, if the $i^{\text{th}}$ node and the $j^{\text{th}}$ node are close to each other on the graph, $\mathbf{e}_i$ and $\mathbf{e}_j$ should be close to each other in the continuous space and vice versa. Interestingly, a graph embedding method can perfectly provide the location representation vectors $\{\mathbf{e}_i\}_{i=1}^n$. Graph embedding maps each node to a vector in the embedding space such that the distance between two points in the embedding space is proportional to the distance of the corresponding nodes on the graph. Accordingly, in order to make the extracted local feature vectors a function of the role of the nodes in the structure of the graph, we propose the approach depicted by the red box in Figure 1. The matrix $\mathbf{F} \in \mathbb{R}^{n \times d_f}$ represents the node labels/attributes and $\mathbf{E} \in \mathbb{R}^{n \times d_e}$ denotes the matrix of the embedding vectors obtained by a graph embedding method (e.g., DeepWalk). The red box represents a GNN Equipped with the Spatial Representation (GNN-ESR). The GNN-ESR leverages both the node labels/attributes and the node embedding vectors to extract the local features. Accordingly, the GNN-ESR is aware of the locations of the nodes and the distance between different nodes and it does not map nodes whose corresponding local structures are similar to the same feature vector. The convolution layers of the GNN-ESR provides a spatial distribution of the local feature vectors which the next layers use to infer the structure of the graph. For instance, in the example discussed in Section 3, the spatial distribution of the local feature vectors corresponding to two graphs in two different classes were indistinguishable. In sharp contrast, if the local feature vectors are extracted using the GNN-ESR, they can build two/three clusters in the feature space if the graph is composed of two/three clusters. Accordingly, the distribution of the feature vectors makes the two classes distinguishable.

---

**Algorithm 1** Node Sampling Using the Geometrical Description of the Graph

---

**Input.** The matrix of embedding vectors $\mathbf{E} \in \mathbb{R}^{n \times d_e}$ and $m$ as the number of sampled nodes.

**0. Initialization.** Sample index $k$ from set $\{i\}_{i=1}^n$ randomly and initialize set $\mathcal{I}_s = \{k\}$.

**1. Repeat $m$ times:** Sample the next embedding vector such that it has the maximum distance to the sampled embedding vectors, i.e., append $k$ to set $\mathcal{I}_s$ such that

$$k = \arg \max_t \left( \min \left( \{ \| \mathbf{e}_t - \mathbf{e}_j \|_2 \}_{j \in \mathcal{I}_s} \right) \right) . \tag{2}$$

**2. Output:** $\mathcal{I}_s$ contains indexes of the sampled nodes.

---

### 4.1 GRAPH POOLING VIA DATA POINT SAMPLING

One of the main challenges of extending the architecture of the CNNs to graphs is to define a pooling function which is applicable to graphs. In this section, the proposed graph pooling method is presented which utilizes the geometrical representation of the graph. The proposed method is composed of two main steps: node sampling and graph down-sampling.

**Node Sampling:** It is not straightforward to measure how accurate a sub-sampled graph represents the topological structure of the given graph. Graph embedding encodes the topological structures of the graph in the spatial distribution of the embedding vectors. Accordingly, the node sampling problem can be simplified into a data point sampling problem. Since the distribution of the embedding vectors represents the topological structure of the graph, we define the primary aim of the proposed pooling function to preserve the spatial distribution of the embedding vectors.

Data point sampling is a well-know problem in big data analysis (Halko et al., 2011) known as column/row sampling problem. A simple method is to perform random data point sampling. However, if the distribution of the data points is sparse in some region of the space, random sampling might not be able to capture the spatial distribution of the data points. Most of the existing sampling methods aim at finding a small set of informative data points whose span is equal to the row-space of the data. However, preserving the row-space of the data is not necessarily equivalent to preserving the spatial distribution of the rows (Rahmani & Atia, 2017). Define $\{\mathbf{e}_i\}_{i \in \mathcal{I}_s}$ as the set of sampled embedding vectors where $\mathcal{I}_s$ is the set of the indexes of the sampled embedding vectors. We define the objective of the sampling method as

$$\min_{\mathcal{I}_s} \sum_{k=1}^n \left( \min \{ \| \mathbf{e}_k - \mathbf{e}_i \|_p \}_{i \in \mathcal{I}_s} \right) \quad \text{subject to} \quad |\mathcal{I}_s| = m , \tag{3}$$

where $|\mathcal{I}_s|$ is the cardinality of $\mathcal{I}_s$. The minimization problem (3) samples $m$ embedding vectors such that the summation of the distances between the embedding vectors and their nearest sampled embedding vector is minimum. This minimization problem is non-convex and it is hard to solve. We propose Algorithm 1 which uses the farthest data point sampling and provides a greedy method to find the sampled data points. Algorithm 1 samples the first embedding vector randomly. In the subsequent steps, the next embedding vector is sampled such that it has the maximum distance to the previously sampled embedding vectors. Accordingly, in each step the embedding vectors which are not close to sampled ones are targeted and gradually the sampled embedding vectors cover the distribution of all the embedding vectors. The computation complexity of Algorithm 1 is $\mathcal{O}(m\,n\,d_e)$.

Although Algorithm 1 starts from a randomly chosen embedding vector, this random choice does not cause variance at test time. Algorithm 1 uses farthest data point sampling and the sampled embedding vectors finally cover the spatial distribution of all the embedding vectors (independent from the starting point). Moreover, one can choose the first embedding vector in a deterministic way. For instance, one candidate could be the embedding vector which is the closest vector to the mean of all the embedding vectors.

**Graph Down Sampling:** Define $\mathbf{X} \in \mathbb{R}^{n \times d}$ as the matrix of feature vectors (Figure 1) and define $\mathbf{X}_s$ and $\mathbf{A}_s$ as the matrix of feature vectors and the adjacency matrix of the sub-sampled graph. The sub-sampled adjacency matrix is obtained via sampling the columns and rows of $\mathbf{A}$ corresponding to the indexes in $\mathcal{I}_s$. We present two methods to obtain $\mathbf{X}_s$.
*Method 1:* Sample the rows of $\mathbf{X}$ corresponding to the indexes in $\mathcal{I}_s$. In other word,

$$\mathbf{X}_s = \mathbf{X}_{\mathcal{I}_s}. \tag{4}$$

*Method 2:* In (4), the feature vectors of the unsampled nodes are discarded. Thus, some useful information could be lost during the down-sampling step. In the second method, we utilize the rows of $\mathbf{X}$ corresponding to the sampled indexes as pivotal feature vectors and all the feature vectors are aggregated around them using the attention technique (Xu et al., 2015). Define $\mathbf{P} \in \mathbb{R}^{m \times d}$ as the matrix of pivotal feature vectors equal to $\mathbf{P} = \mathbf{X}_{\mathcal{I}_s}$ which is equivalent to the matrix of the sub-sampled feature vectors in Method 1. Using the pivotal feature vectors, we define $m$ assignment vectors $\{\mathbf{q}_i \in \mathbb{R}^n\}_{i=1}^m$ as follows

$$\mathbf{q}_i = \text{softmax}(\mathbf{X}\,\mathbf{p}_i^T)\,, \tag{5}$$

where $\mathbf{p}_i$ is the $i^{\text{th}}$ row of $\mathbf{P}$. The vector $\mathbf{q}_i$ is an attention vector which represents the resemblance between all the feature vectors and the $i^{\text{th}}$ pivotal vector. The attention vector $\mathbf{q}_i$ is used to aggregate the feature vectors of $\mathbf{X}$ and obtain the $i^{\text{th}}$ feature vector of the sub-sampled graph as follows

$$\mathbf{x}_{s\,i} = \mathbf{q}_i^T\mathbf{X}\,, \tag{6}$$

where $\mathbf{x}_{s\,i}$ is the $i^{\text{th}}$ row of $\mathbf{X}_s$.

**Remark 1.** *If the nodes of the given graph are sparsely connected, the down-sampled graph can be a disconnected graph. Define $\mathbf{A}_r = \sum_{i=1}^r \mathbf{A}^i$. If the distance between the $j^{th}$ node and the $k^{th}$ node is less than $r+1$, $\mathbf{A}_r(j,k)$ is non-zero. Therefore, if we down-sample $\mathbf{A}_r$ to obtain the adjacency matrix of the sub-sampled graph, two sampled nodes are connected if their distance is less than $r+1$. In the presented experiments, we down-sized $\mathbf{A}^3$ to obtain $\mathbf{A}_s$. In most of the graph classification applications, the size of the graphs is not large. However, if in a specific application the size of the graphs is large, the computation of $\mathbf{A}^k$ (for $k \geq 2$) could require large volume of memory. In this scenario, one can simply down sample the adjacency matrix $\mathbf{A}$ and find the disconnected nodes and subsequently connect each disconnected node to its nearest sampled node (nearest according to the original graph). In addition, if we use $\mathbf{A}^k$ to obtain the down sampled adjacency matrix and the obtained adjacency matrix is dense, one can sparsify the matrix by applying a proper threshold to the values of the down sampled adjacency matrix. In the presented experiments, we did not face this problem and we used the exact down sampled version of $\mathbf{A}^3$.*

**Remark 2.** *We presented two methods to obtain $\mathbf{X}_s$. In the presented experiments, we used both of them to obtain $\mathbf{X}_s$. The final $\mathbf{X}_s$ was obtained as a convex combination of them.*

## 5 NUMERICAL EXPERIMENTS

First we focus on demonstrating the significance of equipping the GNN with the spatial representation. Subsequently, the proposed pooling method (Spatial Pooling) is compared with the existing graph pooling methods.

Table 2: Classification accuracy when the node labels are included.

|  | PTC | PROTEINS | DD | ENZYM | SYNTHIE | NLC | MDC |
|---|---|---|---|---|---|---|---|
| GNN | 77.44 ±6.33 | 80.01 ±3.57 | 82.90 ±2.70 | 61.66 ±4.68 | 67.75 ±3.436 | 90.33 ±2.17 | 86.44 ±2.94 |
| GNN-ESR | 77.64 ±6.89 | 80.54 ±3.17 | 83.33 ±3.18 | 69.16 ±4.54 | 71.15 ±2.54 | 100 ±0 | 97.50 ±1.64 |
| Improvement % | 0.25 | 0.66 | 0.51 | 12.17 | 5.01 | 10.70 | 12.80 |

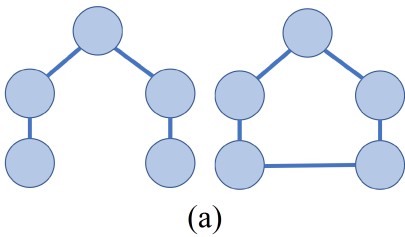 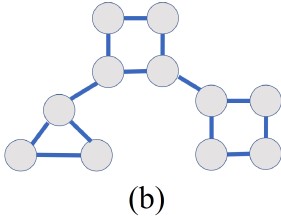

(a)  (b)

Figure 2: **Part (a)**: This part represents two graphs. Each circle represents a cluster of nodes. The clusters in the right graph form a loop but in the left graph they do not. **Part (b)**: Each circle represents a cluster of nodes. This graph is composed of 11 clusters and the clusters form 3 loops. The task is to count the number of loops made by the clusters.

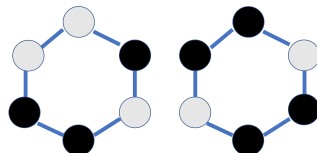

Figure 3: Each circle represents a cluster and both graphs are composed of 6 clusters. A white cluster means that the labels of its corresponding nodes are equal to 1 and black means they are labeled 0. In the NLC task, the neural network is trained to count the number of white clusters. In the MDC, the neural network is trained to check if the diagonal clusters are labeled similarly. A pair of clusters are called diagonal clusters if they are in the maximum distance from each other. In the left graph, the labels of one pair of diagonal clusters are not equal. In the right graphs, the labels of all the diagonal clusters are equal.

**The structure of the basis neural network:** In the presented experiments, GNN-ESR indicates the neural network depicted by the red box in Figure 1 with $k_1 = 1$ and $k_2 = 2$. The dimensionality of the output of all the convolution layers is equal to 64. The functionality of each convolution layer can be written as (1) where $f(\cdot)$ is the Relu function followed by Batch-normalization (Ioffe & Szegedy, 2015). The output of all the convolution layers are concatenated (Xu et al., 2018) to obtain the global representation of the graph and the element-wise max function is used as the global aggregator. The fully connected classifier is composed of three fully connected layers. The first two layers of the fully connected classifier are equipped with dropout (Srivastava et al., 2014) ( dropout probability equal to = 0.5) and batch-normalization. In the presented tables, GNN indicates a graph neural network similar to the GNN-ESR while $\mathbf{E}$ is not provided as an input.

**The input and the optimizer:** The Deep-Walk graph embedding method (Perozzi et al., 2014) was used to embed the graphs and the dimension of the embedding vectors is equal to 12. The length of the random walks is determined as $\max(4, \min(n/10, 10))$ where $n$ is the number of the nodes. All the neural networks were similarly trained using the Adam optimizer (Kingma & Ba, 2015). The learning rate was initialized at 0.005 and was reduced to 0.0005 during the training process. Following the conventional settings, we perform 10-fold cross validation (Zhang et al., 2018). We refer the reader to (Kersting et al., 2016) for a complete description of the used real-world datasets. The size of each synthetic dataset is equal to 1000.

## 5.1 Analyzing unlabeled/non-attributed graphs

In this section, we consider graphs whose nodes are not labeled/attributed. In some of the utilized real datasets, the nodes are labeled/attributed which are discarded in this experiment to examine the ability of the neural networks in inferring the topological features of the graphs in the absence of the labels/attributes. First we describe the designed synthetic graph classification tasks.

*High Level Loop Detection (**HLLD**):* In this task, the generated graphs are composed of 3 to 6 clusters (the number of clusters are chosen randomly per graph). Each cluster is composed of 20 to 45 nodes (the number of nodes are chosen randomly per cluster). Each node in a cluster is connected to 5 other nodes in the same cluster. In addition, if two clusters are connected, 3 nodes of one cluster are densely connected to 3 nodes of the other cluster. In the generated graphs, the consecutive clusters are connected. The classifiers are trained to detect if the clusters in a graph form a loop. Obviously, there are many small loops inside each cluster. The task is to detect if the clusters form a high level loop. Accordingly, this task is equivalent to a binary classification problem. Part (a) of Figure 2 shows an example of the HLLD task.

*Count Number of Clusters (**CNC**):* In this task, the graphs are similar to the graphs in the HLLD task. However, the objective of the CNC task is to count the number of clusters. The generated graphs are composed of 3 to 6 clusters. This task is equivalent to a classification task with four classes.

*Count the Number of the Loops of Clusters (**CNLC**):* In this task, the generated graphs contain several clusters and the clusters form multiple loops. For instance, the graph depicted in Part (b) of Figure 2 consists of 11 clusters and the clusters form form 3 loops. In this experiment, the clusters can form 2 to 4 loops and each loop is formed with 3 to 5 clusters. The number of clusters in each loop is chosen randomly per loop. The objective of this task is to count the number of the loops. Thus, this task is equivalent to a graph classification task with three classes.

Table 1 shows the classification accuracy of a simple GNN and the GNN-ESR. The last three columns show the performance with the synthetic datasets. One can observe that the GNN failed to learn to perform the synthetic tasks. The main reason is that in the synthetic tasks, the local structure corresponding to all the nodes are similar and the convolution layers map all the nodes to similar vectors in the feature space. Thus, the GNN cannot extract discriminative features from the distribution of the local feature vectors. In sharp contrast, the GNN-ESR is aware of the differences/similarities between the nodes and it does not necessarily map them to similar feature vectors when the nodes are not close to each other on the graph. The results show that the GNN-ESR successfully extracts discriminative features from the distribution of the extracted local feature vectors. Moreover, the results with real datasets demonstrate that the proposed approach significantly (up to 23 %) improves the classification accuracy. The spatial representation makes the GNN-ESR aware of the difference between the nodes (although they are not labeled) and it paves the way for the GNN-ESR to extract discriminative features from the distribution of the embedding vectors.

## 5.2 Analyzing labeled graphs

This experiment is similar to the previous experiment but the node labels are included. We use the real datasets and the following synthetic tasks.

*Number of Labeled Clusters (**NLC**):* The graphs in this task are similar to the graphs in the CNC task. Each graph is composed of 6, 7, or 8 clusters. The consecutive clusters are connected and they form a loop. The labels of the nodes are binary (0 or 1) and all the nodes in the same cluster are labeled similarly. The task is to count the number of clusters whose nodes are labeled 1. There are 4 classes of graphs, i.e., the number of clusters which are labeled 1 are equal to 2, 3, 4, or 5. Figure 3 shows an example of this task.

*Match the Diagonal Clusters **MDC**:* In this task, each graph is composed of 6, 8, or 10 clusters and the clusters form a loop. The labels of the nodes are binary and all the nodes in a cluster are labeled similarly. The task is to check if all the pairs of diagonal clusters are labeled similarly. A pair of clusters are called a pair of diagonal clusters if they are in the maximum distance from each other. Therefore, the MDC task is equivalent to a binary classification task. Figure 3 shows an example of this graph classification task.

Table 2 shows classification accuracy with different datasets. One can observe that even when the nodes are labeled, the difference between the performances is significant for most of the datasets. The main reason is that the spatial representation makes the neural network aware of the differences between the nodes and the neural network leverages both the node labels and the node embedding vectors to extract discriminative features.

## 5.3 GRAPH POOLING

In this experiment, the proposed graph pooling method (Spatial Pooling) is compared with some of the recently published pooling algorithms. In all the neural networks, the described GNN-ESR is used to extract the local feature vectors. Similar to the architecture depicted in Figure 1, the pooling layer is placed next to last convolution layer of the GNN-ESR. Since the graphs in the datasets are not large graphs (mostly less than 100 nodes), one down-sampling layer was used.

• Spatial Pooling: Two spatial convolution layers were implemented after the down-sampling step. Define $\mathbf{Y} \in \mathbb{R}^{n \times 192}$ as the concatenation of the outputs of all the convolution layers before the pooling layer (three convolution layers) and define $\mathbf{y} \in \mathbb{R}^{192}$ as the aggregation of these $n$ feature vectors (by the element-wise max function). Similarly, define $\mathbf{Y}_s \in \mathbb{R}^{m \times 128}$ as the concatenation of the outputs of all the convolution layers after the pooling layer and define $\mathbf{y}_s \in \mathbb{R}^{128}$ similar to $\mathbf{y}$. The final representation of the graph is defined as the concatenation of $\mathbf{y}$ and $\mathbf{y}_s$ and it is used as the input of the final classifier. We chose $m = \min(30, \max(5, n/4))$.

• Diff-Pool (Ying et al., 2018): This method was implemented according to the instructions in (Ying et al., 2018). Two spatial convolution layers were placed after the down-sizing step. The final representation of the graph was obtained similar to the procedure used in Spatial-Pooling.
• Rank-PooL (Gao & Ji, 2019; Lee et al., 2019): Similar to the other pooling methods, one down-sampling layer was used and two spatial convolution layers were implemented after the down-sizing step. The number of sampled nodes was chosen similar to the proposed method.
• Sort-Node (Zhang et al., 2018): Similar to the implementation described in (Zhang et al., 2018) and its corresponding code, 30 nodes were sampled to construct the ordered sequence of the nodes.

Table 3: Classification accuracy of the GNN-ESR with different pooling methods.

|  | MDC | PROTEINS | DD | ENZYM | SYNTHIE |
|---|---|---|---|---|---|
| Global Max-Pooling | 97.50 ±1.64 | 80.54 ±3.17 | 83.33 ±3.18 | 69.16 ±4.54 | 71.15 ±2.54 |
| Spatial-Pooling | **98.33** ±0.66 | **80.72** ±3.42 | 83.52 ±2.35 | **71.33** ±4.42 | **72** ±2.87 |
| Sort-Pooling | 70 ±5.25 | 80.54 ±4.56 | 82.39 ±2.51 | 63.16 ±3.68 | 67.75 ±3.05 |
| Diff-Pool | 97.8 ±1.46 | 80.45 ±3.36 | **84.36** ±2.66 | 70.33 ±5.26 | 71.75 ±3.36 |
| Rank-PooL | 97.66 ±1.23 | **80.72** ±3.12 | 83.67 ±3.83 | 67.5 ±7.68 | 71.5 ±2.78 |

Table 3 shows the accuracy of the GNN-ESR with different pooling methods. One can observe that Spatial-Pooling achieves higher or comparable results on all the datasets. On the MDC dataset, Spatial-Pooling outperforms the other methods and on ENZYM and SYNTHIE datasets, it slightly outperforms Diff-Pool. The Diff-Pool method also achieves notable performance on most of the datasets. The feature vectors which Diff-Pool uses to down-size the graph are fixed. In contrast, the way the proposed approach down-sizes the graph depends on the topology of the graph not a set of fixed cluster centers. One can observe that the accuracy of Sort-Pool is significantly less than the accuracy of Spatial-Pooling and Diff-Pool with the MDC dataset. The main reason is that Sort-Pool sorts the nodes in a 1-dimensional array. The placement of the nodes in a 1-dimensional array leads to loosing important information about the structure of the graph. An observation one can make by studying the results is that the performance of the GNN-ESR is close to the performance of the GNN-ESR with the pooling layers. In contrast, the performance of a simple GNN with the pooling layers can be notably higher than the performance of the GNN (Yanardag & Vishwanathan, 2015). For instance, Table 4 shows the performance of the neural networks with and without the pooling layers. One can observe that the performance of the GNN equipped with the pooling layer is notably higher than the performance of the GNN on the MDC and the ENZYM datasets. In contrast, the difference between the performances reported in the third and the forth rows is lower. Another predictable point that Tables 4 makes is that the pooling layer cannot make the GNN able to learn

to perform the CNC task or the HLLD task (and also the CNLC task). In these tasks, the neural network should be able to distinguish the nodes to be able to infer the structure of the graph.

Table 4: Classification accuracy of the neural networks with and without the pooling layer.

|  | MDC | CNC | HLLD | ENZYM | SYNTHIE |
|---|---|---|---|---|---|
| GNN | 86.44 | 36.40 | 54.24 | 61.66 | 67.75 |
| GNN + Diff-Pool | 95.55 | 36.51 | 56.33 | 69.49 | 67 |
| GNN-ESR | 97.50 | 99 | 99.10 | 69.16 | 71.15 |
| GNN-ESR + Diff-Pool | 97.8 | 100 | 100 | 70.33 | 71.75 |

## 6 CONCLUSION

An important shortcoming of the GNN was discussed . It was shown that if the nodes/edges of the given graph are not labeled/attributed or the labels/attributes do not make the GNN aware of the role of the nodes in the structure of the graph, the GNN can fail to infer the topological structure of the graph. Motivated by the success of the deep networks in analyzing point-cloud data, we proposed an approach in which a geometrical representation of the graph is provided to the GNN. The geometrical representation is computed by a graph embedding method which encodes the topological structure of the graph into the spatial distribution of the embedding vectors. We showed that the proposed approach significantly empowers the GNN via analyzing its performance on a diverse set of real and synthetic datasets. Moreover, the spatial representation was utilized to simplify the graph down-sampling problem and a novel graph pooling method was proposed.

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
