# OpenReview forum: "GRAPH ANALYSIS AND GRAPH POOLING IN THE SPATIAL DOMAIN"
_ICLR.cc/2020/Conference — Reject_

### Official Review · AnonReviewer1 · 2019-10-23
**Official Blind Review #1**

**Rating:** 3

**Review:**

The authors propose in this paper to complement the node attributes in a graph with vectors obtained using a graph embedding algorithm. More precisely, they propose a graph neural network that apply several layers of graph convolution in parallel to the node attributes and to the embedding, then takes an average of the result, which is fed to another series of graph convolution. This is combined with some form of sampling which strongly resembles median based quantization but is solved with some basic heuristics and without acknowledging the resemblance (I might be missing something).

Globally, I find this paper very unclear. The authors are mixing references to graph convolution for fixed graph structure with references about general graph neural networks. Their goal is not clearly stated from the beginning and in many places the discussion lacks of focus and is difficult to follow. In the experimental evaluation several values are very close one to another and the statistical significativity of the differences is not assessed.

**Experience Assessment:**

I have read many papers in this area.

**Review Assessment: Checking Correctness Of Derivations And Theory:**

I did not assess the derivations or theory.

**Review Assessment: Checking Correctness Of Experiments:**

I assessed the sensibility of the experiments.

**Review Assessment: Thoroughness In Paper Reading:**

I read the paper at least twice and used my best judgement in assessing the paper.

---

> ### Author Response · Authors · 2019-11-14
> **Response**
>
> We thank the reviewer for the comments.
>
> 1- “The authors are mixing references to graph convolution for fixed graph structure with references about general graph neural networks.”
> We are not sure if we understood this comment well. But, the spatial graph convolution used in all the scenarios is similar. Specifically, we used spatial convolution expressed in eq (1) as our convolution function.
>
> 2- “their goal is not clearly stated - I find this paper very unclear”:
> In abstract and in section 3, the goal of the paper is explained with illustrative examples. In addition, several novel experiments are presented which showcase the importance of the proposed approach (Table 1 and Table 2).
>
>
>
> 3- “In the experimental evaluation several values are very close one to another”
> In Table 1 and Table 2, it is shown that the improvement in the performance of the GNN is significant when the proposed approach is utilized.

---

> > ### Comment · AnonReviewer1 · 2019-11-15
> > **Significance**
> >
> > You write "In Table 1 and Table 2, it is shown that the improvement in the performance of the GNN is significant when the proposed approach is utilized. ". However for instance in Table on PTC, the differences between GNN and GNN-ESR cannot be statistically significant, according e.g. to a Welch's t test. You should include statistical test results.

---

> > > ### Author Response · Authors · 2019-11-15
> > > **response**
> > >
> > > We thank the reviewer for replying to our answers.
> > >
> > > First, we would like to note that we included error bars in the new paper as it was requested by reviewer 4. Second, please let us clarify what we wanted to show in Table 1 and Table 2. GNN-ESR yields notable performance gain when:
> > >
> > > 1-    The graphs are composed of similar local structures and the difference between graphs is in their global structure. For instance, Table 1 shows that for the HLLD, CNLC, and CNC tasks, GNN-ESR yields notably higher performance (around 100 % performance improvement).
> > >
> > > 2-    When the nodes/edges of the graph are not labeled or attributed. In Table 1, we did not include the node labels and results for PROTEINS, NCI1, DD, ENZYM, SYNTHIE shows that the proposed approach clearly helps the GNN to extract more informative features.
> > >
> > > PLEASE NOTE that we could include PTC in this Table 1 too and the performance gain for the PTC dataset when the nodes are not labeled is notable (6 %). Table 1 was too long and we did not include further datasets.
> > >
> > > 3-    Table 2 shows that even if the nodes/edges of the graphs are labeled/attributed, the proposed approach can provide notable performance gain and the reason is twofold. First of all, the proposed approach helps the GNN to distinguish similar local structures in different locations of the graph. Second, it provides a geometrical representation of the graph to the GNN. For instance, for the ENZYM, NLC, and MDC, the performance gain is more than 10 %. We intentionally included PTC, PROTEINS, DD in Table 2 to show that the proposed approach either improves the performance or yields comparable performance.
> > >
> > > In the paper, we claim that the proposed approach CAN improve the results. The amount of improvement depends on the structure of the graphs and the way the graphs are attributed/labeled.
> > >
> > > We would also like to note that the proposed approach also addresses the problem raised by (Xu et al. ICLR 2019) in a more effective way. The authors of (Xu et al. (ICLR 2019)) wanted to make sure that if two local structures are different, the GNN maps them to different vectors in the feature space. The solution that we proposed prevents the GNN from mapping even similar local structures to the same feature vectors because we make the GNN aware of the location of the nodes. Please take a look at our answer to Reviewer 4 about this point.
> > >
> > > We also like to mention that the contribution of our paper is not limited to GNN-ESR. We utilized the spatial representation of the graph to propose a novel pooling method too. The proposed approach is more intuitive and interpretable than most of the existing methods and the results show that it either outperforms or yields comparable performance.

---

### Official Review · AnonReviewer2 · 2019-10-24
**Official Blind Review #2**

**Rating:** 6

**Review:**

In this work the authors point out an issue related to graph neural networks. Specifically, if two nodes, that may be far apart in the graph, may be represented as (almost) the same vector. This is simply because when no features/labels are associated with nodes, and the local structure around those two nodes is very similar then the local aggregation of information will result in a similar representation.  Therefore the authors introduce an embedding first of the graph in the Euclidean space using DeepWalk and then use this embedding in combination with the design of a CNN. The authors propose a pooling method that outperforms several state-of-the-art pooling techniques on real data. Overall, the empirical results are supportive of the fact that the proposed method can help improve the performance of GNNs.

Overall I found the results of this paper to be weak, but nonetheless the paper is well-written and contains some interesting ideas. Hence my rating. Some questions follow.

- While the authors call this as an "issue" it is more like a feature of these methods.  For instance, in "RolX: Structural Role Extraction & Mining in Large Graphs" by Henderson et al. this "issue" could turn out to be an interesting feature of the GNNs in the sense that these nodes may have a similar (structural) role.  It would be nice to have a short discussion related to this line of research in social networks' analysis.
-  Some components of the CNN (e.g., node sampling) could be done using  well-developed tools for sampling matrices from numerical linear algebra, or by introducing some randomness when sampling a node as in kmeans++.
- Graph downsampling appears to be an expensive operation. Can you please comment on the running times? The issue of scalability is not discussed, and the reader cannot easily infer to what sizes this method can scale up to.
- Using other graph tasks, that are classical but also more challenging (e.g., learning 2-connected components of a graph just to mention something that comes up) would be interesting to see in Section 5.2.
-  It would have been interesting to see the effect of the embedding step on the accuracy on the real data. E.g., using node2vec or standard spectral embeddings.


[Edit: The authors have replied to my comments, and the other reviewers' comments in great detail. Therefore I upgrade my score.]

**Experience Assessment:**

I have published in this field for several years.

**Review Assessment: Checking Correctness Of Derivations And Theory:**

I assessed the sensibility of the derivations and theory.

**Review Assessment: Checking Correctness Of Experiments:**

I assessed the sensibility of the experiments.

**Review Assessment: Thoroughness In Paper Reading:**

I read the paper thoroughly.

---

> ### Author Response · Authors · 2019-11-14
> **Response**
>
> We thank the reviewer for the comments.
>
> 1- “Overall I found the results of this paper to be weak”:
>
> Table 1 and Table 2  indicate that the improvements in the results are significant for both real and synthetic data.
>
> 2- “Some components of the CNN (e.g., node sampling) could be done using  well-developed tools for sampling matrices from numerical linear algebra, or by introducing some randomness when sampling a node as in kmeans++”:
>
> Thanks for this suggestion. We are well aware of column/row sampling techniques and we considered several sampling methods as our candidates including the one used in K-means++ and SRS [arXiv:1705.03566]. The farthest data point sampling method used in our pooling method (which is very similar to the sampling techniques used in K-means ++) ensures that the sampled embedding vectors cover the spatial distribution of all the embedding vectors and this is all we expect the sampling method to do.
>
> 3- “Graph downsampling appears to be an expensive operation.”
> The complexity of the sampling method is linear with the number of sampled embedding vectors (O(m*n*de) where m is the number of sampled vectors and de is the dimension of the embedding vectors). In addition, different from node classification, in the graph classification task, we mostly do not have large graphs (mostly less than 100 nodes). In the revised paper, we described the computation complexity of the node sampling method.
>
> 4- “It would have been interesting to see the effect of the embedding step on the accuracy on the real data. E.g., using node2vec or standard spectral embeddings.”
>
> ّIn our initial experiments, we used other embedding methods such as [arXiv:1710.02971] but the results were not much different. In the final version, we will report the results with several different embedding methods.
>
> 5- “While the authors call this as an "issue" it is more like a feature of these methods.  For instance, in "RolX: Structural Role Extraction & Mining in Large Graphs" by Henderson et al. this "issue" could turn out to be an interesting feature of the GNNs in the sense that these nodes may have a similar (structural) role.”
>
> It depends on the application that we deal with. In the graph classification problem , we clearly showed that it is not a desirable feature and this feature can make the GNN unable to extract discriminative features.
> Moreover, our approach addresses another problem of GNNs raised by (Xu et al. (ICLR 2019)): GNNs can map even different local structures to same feature vectors.
> Since we include the information about the location of the nodes, we help the GNN to distinguish both similar and different local structures.
> We cited “RolX: Structural Role Extraction & Mining in Large Graphs” and clarified that the point that in some applications mapping node with similar local structures to similar feature vectors can be desirable.
>
> We would like to reiterate that our paper is the first work which address this problem and the presented results (Table 1 and Table2) clearly show how effective is the presented solution.

---

### Official Review · AnonReviewer4 · 2019-10-31
**Official Blind Review #4**

**Rating:** 6

**Review:**

============ After rebuttal ============
I thank the authors for carefully discussing the points of my review. I have upgraded the score to marginal acceptance (6).

============ Original review ============

In this paper, the authors identify a shortcoming of existing GNN architectures for graph classification tasks -- specifically, the fact that, in the featureless regime, the graph convolutional layers rely on propagating very rudimentary structural information, making it hard (or impossible) to distinguish graphs with similar local structure. To fix the problem, the authors propose to augment the input feature space with graph-structural embeddings (computed by an algorithm like DeepWalk), and processing those in parallel with any other input features available.

On existing real-world datasets, as well as synthetic datasets carefully constructed to illustrate this phenomenon, the proposed pipeline matches or exceeds the version without the structural embedding inputs. Further, the authors note that the structural embeddings could be used to propose a novel graph pooling method -- one which attempts to preserve as diverse structural feature sets as possible. It is shown that this method is competitive to other differentiable pooling methods, like DiffPool and Graph U-Nets. Lastly, the authors demonstrate that the addition of pooling layers does not help baseline GNNs on the synthetically constructed tasks, as the fundamental issue of handling similar local structures is still not addressed.

I believe that the paper clearly exposes and proposes a nice idea which could hold great potential, and which can be useful to graph representation learning practitioners. I am particularly happy with the design of the synthetic experiments. However, I find that, in its current form, the manuscript is narrowly below the bar for a venue like ICLR.

Comments:
* The observation that existing GNN layers may struggle with distinguishing featureless graphs is not particularly novel. It's largely the centerpiece of the (already cited) work of Xu et al. (ICLR 2019), and I believe that its relevance and relation to the authors' work should be better stressed in the related work section.

* The usage of DeepWalk to encode structural information (and even to be used as initial features for a GNN) is, ultimately, also not necessarily a novel idea. At least, it's something that should be clear to any expert GNN practitioner already: if useful features are missing from the graph, a method like DeepWalk (if applicable; see below!) could be a go-to method for obtaining such features. In its current form, I don't see that the authors are proposing anything substantially architecturally novel, and their contribution is primarily on the data/feature engineering side.

* The above point is not necessarily problematic, but if the aim is to stress the importance of the architectural novelties of the proposed GNN-ESR model more, and not just the added features, I would recommend the authors to perform a few ablations: e.g. seeing how well would processing a concatenation of E and F in the same GNN layer perform.

* Many of the standard graph classification datasets are known to be noisy and unreliable (see e.g. Luzhnica, Day and Liò (ICML GraphReasoning Workshop 2019). This means that it is a must to report error bars of the cross-validation experiments. It's hard to say that many of the improvements depicted here are statistically significant otherwise.

* I have concerns about the computational complexity, or even feasibility, of using DeepWalk-like methods in the general case, e.g. for node classification. Namely, if such layers are to be applied in inductive settings (with nodes gradually added to graphs), one would require re-running DeepWalk every time a new node is added. The authors should comment on this adequately, and perhaps discuss the feasibility of other unsupervised embedding techniques for obtaining the e-vectors -- such as VGAE (Kipf and Welling, NIPS BDL 2016), GraphSAGE (Hamilton et al., NIPS 2017), Graph2Gauss (Bojchevski and Günnemann, ICLR 2018) or DGI (Veličković et al., ICLR 2019).

* While I find the proposed pooling method interesting (and more grounded in the graph's structural features than other proposed works), I find that there are many potential limitations to be discussed. For example, the fact that we start from a random first index means that we cannot rely on the obtained pooling to always be the same -- could this cause undesirable variance at test time? Furthermore, the downsampling from A^3 is a sure-fire way to obtain dense graphs after the first pooling -- potentially severely limiting the applicability of the method for large graphs. In my opinion, the authors should appropriately comment on these and perform ablations against pooling with A and A^2 (as was done in Graph U-Nets). It should also be interesting to note that there exist other structurally-informed pooling methods; see e.g. the Clique pooling method from Luzhnica, Day and Liò (ICLR RLGM 2019).

Given all of the above, I recommend (marginal) rejection, but am open to improving my score if the authors appropriately address the aforementioned comments.

Some minor comments and thoughts:
* The paper has several typos and grammar issues, and a typo pass is highly encouraged to aid clarity;
* The "attention mechanism" of Equation (5--6) seems to be nonparametric? If so, it might be interesting to compare with a version that features learnable queries, e.g. using the Transformer-style attention.
* In Equation (3), should the first min actually be a max? As we're maximising the overall minimum distances between topological features.
* I'm not sure that the paper is anywhere clear on what's the exact GNN layer being used. Could this be clarified and made more explicit? It's critical to reproducibility.
* I find it curious that the authors needed to resort to using batch normalisation---I usually found it to either have no meaningful effect on the results on the graph classification benchmarks, or it made results worse. Can the authors comment on this decision?
* The idea to concatenate output of all convolutional layers is heavily resembling Jumping Knowledge networks (Xu et al., ICML 2018), and I believe they should be appropriately cited.

**Experience Assessment:**

I have published in this field for several years.

**Review Assessment: Checking Correctness Of Derivations And Theory:**

N/A

**Review Assessment: Checking Correctness Of Experiments:**

I carefully checked the experiments.

**Review Assessment: Thoroughness In Paper Reading:**

I read the paper at least twice and used my best judgement in assessing the paper.

---

> ### Author Response · Authors · 2019-11-14
> **Response**
>
> We thank the reviewer for the comments.
>
> 1- “Xu et al. (ICLR 2019)”:
> Please note that the approach presented in (Xu et al. (ICLR 2019)) does not solve the problem that we address in our paper. The authors of (Xu et al. (ICLR 2019)) want to make sure that if two local structures are DIFFERENT, the GNN maps them to different vectors in the feature space and they showed that the sum aggregator satisfies this requirement. The solution that we proposed prevents the GNN from mapping even SIMILAR local structures to the same feature vectors because we make the GNN aware of the location of the nodes. In addition, in all our experiments we used the sum aggregator. The presented experiments clearly show that  (Xu et al. (ICLR 2019))  can not solve the problem that we addressed in our paper.
>
> We clarified this difference in the new paper.
>
> 2- “their contribution is primarily on the data/feature engineering side”:
> We believe that it is an important contribution that our paper shows that the fact that GNN maps similar local structures to same feature vectors is problematic. In addition, we propose a solution to this important problem and our solution is simple and effective ( it also addresses the problem raised in Xu et al. (ICLR 2019) in a more effective way).
> Moreover, the contributions of our paper is not limited to GNN-ESR. We present a new graph pooling method too.
>
>
> 3- “The above point is not necessarily problematic but … seeing how well would processing a concatenation of E and F in the same GNN layer perform”:
>
> We do not claim that we propose a novel architecture for GNNs (except the new pooling layer).  In Figure 1, if we set k1=0, it is equivalent to the scenario in which we  E and F are concatenated. Our initial experiments with few data-sets showed that k=1 is slightly better than k=0. We will regenerate the results with k=0 and will report them in the final version.
>
>
> 4- “This means that it is a must to report error bars of the cross-validation experiments..”:
> Thanks for your suggestion.  We added the error bars to the results in the new paper.
>
>
> 5- “computational complexity, or even feasibility, of using DeepWalk-like methods in the general case … or DGI (Veličković et al., ICLR 2019).”
>
> In this paper, we focus on graph classification. For online cases, there are some online embedding methods. In addition, Deepwalk is scalable to large graphs. In the paper, we mention that the motivation for using the embedding vectors is to provide a point-cloud representation of the graph to the neural networks. Thus, any embedding method which yields embedding vectors such that the distance between the embedding vectors is proportional to the distance of their corresponding nodes is applicable. In our initial experiments, we used other embedding methods such as [arXiv:1710.02971] but the results were not much different.
> Please note that methods such as VGAE (Kipf and Welling, NIPS BDL 2016) are not applicable to our method. They basically assume that the node attributes contain sufficient amount of information such that one can recover the adjacency matrix via processing the node attributes.
>
>
> 6- “the fact that we start from a random first index means ... could this cause undesirable variance at test time? ”:
>
> Since we use farthest data point sampling, finally the sampled embedding vectors cover the spatial distribution of all the embedding vectors. In addition, one can start from a deterministic embedding vector. For instance, we can start from the embedding vector which is the closest vector to the mean of all the embedding vectors.
>
> 7- “Furthermore, the downsampling from A^3 is a sure-fire way to obtain dense graphs after the first pooling -- potentially severely limiting the applicability of the method for large graphs”:
>
> Adjacency matrices are mostly sparse and obtaining A^2 or A^3 is not computationally expensive. In addition, in most of the graph classification applications, the size of the graphs are small.
>
>
> 8- “CLIQUE POOLING”:
> Thanks for bringing this paper into our attention. We cited it.
>
> 9- “I recommend (marginal) rejection, but am open to improving my score if ....”
>
> We are delighted that you found our paper innovative and we would like to note that openreview shows “clear reject” not marginal reject. We hope that our answers provide a more clear picture of the contributions of the paper.
>
> 10- “it might be interesting to compare with a version that features learnable queries”:
>
> Both Diffpool and Rank-Pool use learnable queries.
>
>
> 11- “In Equation (3), should the first min actually be a max?”
>
> It should be min. The distance between each data point and its closest sampled vector matter.
>
> 12- “what's the exact GNN layer being used”
>
> We used spatial convolution layer as in eq (1). We will release the data and the code after acceptance.
>
> 13- “Batch normalization”:
>
> We used it to help to prevent overfitting along with dropout.
>
> 14- (Xu et al., ICML 2018):
>
> We cited it. Thanks.

---

> > ### Comment · AnonReviewer4 · 2019-11-14
> > **Thank you**
> >
> > Thank you for your reply, which addresses some of my queries appropriately. I would like to follow up on the points below. If properly addressed, I will increase my score to Weak accept.
> >
> > 4. Thank you for including error bars -- this puts the results in a more clear context. You should make it clear in the paper on which datasets your methods no longer have a clear edge. (This is fine and to be expected -- most of the existing graph classification datasets are not diverse enough to demonstrate differences between models -- and it's much better to be honest to the readers about it.)
> >
> > 5. Thank you for the response. It has been raised in some works (e.g. DGI) that a GNN by itself---irrespective of the parameters chosen---imposes an inductive bias that close nodes should be grouped together. It would be interesting to see how the method performs if you start with randomly-initialised vectors in every node, and then just pass them through a few layers of randomly initialised GNNs to obtain e-vectors. This method is scalable and should capture very similar invariances as DeepWalk.
> >
> > 6. Thank you for your thoughts -- could you please include this discussion in the paper somewhere?
> >
> > 7. I'm not claiming that obtaining A^2 or A^3 is computationally expensive -- I'm merely stating that the matrices will become dense (and maybe even complete, very quickly). This poses issues on the memory side (while I appreciate that the classification tasks you used aren't large-scale, you should at least comment on the fact the method as-is won't be applicable memory-wise for large graphs). It also poses issues on the message aggregation side -- when you need to deal with O(N) GNN messages in every node, you can easily run into problems of oversmoothing or exploding messages. This should all be at least briefly dicussed in the paper.
> >
> > 9. (OpenReview shows "Weak reject" to me, which is the highest possible 'reject' score.)
> >
> > 10. Yes, DiffPool etc. use learnable queries, but they don't pool in the same way as the method proposed here. I was interested in how the exact pooling method you propose would fare if the attention was learnable.
> >
> > 11. I'm not sure I follow the argument here. If both of your operators in eqn (3) are 'min', then why in Algorithm (1) is an argmax performed? If one wanted to minimise the sum of minimal distances (min-of-min, as eqn (3) shows), why not take argmin?
> >
> > 12. Okay, this clarifies my doubt. Could you please make it more clear somewhere in the paper that you're using exactly Equation (1) in your implementation?

---

> > > ### Author Response · Authors · 2019-11-14
> > > **Response**
> > >
> > > We thank the reviewer for the comments and the constructive suggestions.
> > >
> > > 4. “Thank you for including error bars -- this puts the results in a more clear context. You should make it clear in the paper on which datasets your methods no longer have a clear edge.”
> > >
> > > Thanks for your suggestion. In the paper, we did not say that the proposed method outperforms on all the data-sets. We stated that the algorithm outperforms or yields comparable performance. In the revision, it is stated in a more clear way.
> > >
> > > 5. “Thank you for the response. It has been raised in some …  is scalable and should capture very similar invariances as DeepWalk.”
> > > It is an interesting thought and we can consider it in our final version. I think since the GNN averages the nearby nodes, it makes the vectors of two connected nodes close to each other. However, if the final purpose is to create something which is already given by DeepWalk in better quality, why not just using DeepWalk. It is also scalable and its complexity is linear with the number of nodes. In addition, since the GNN starts with random noise, it is hard for the GNN to create an informative distribution of the embedding vectors using just a few convolutions.
> > >
> > >
> > > 6. “Thank you for your thoughts -- could you please include this discussion in the paper somewhere? “
> > >
> > > Thanks for the suggestion. We did (page 6 - right above graph down sampling).
> > >
> > > 7. I'm not claiming that obtaining A^2 or A^3 is computationally expensive -- I'm merely stating that the matrices will become dense (and maybe even complete, very quickly). This poses issues on the memory side (while I appreciate that the classification tasks you used aren't large-scale, you should at least comment on the fact the method as-is won't be applicable memory-wise for large graphs). It also poses issues on the message aggregation side -- when you need to deal with O(N) GNN messages in every node, you can easily run into problems of oversmoothing or exploding messages. This should all be at least briefly discussed in the paper.
> > >
> > > Thanks for the comment. We extended Remark 1 to include more details and some solutions.
> > >
> > > 11. I'm not sure I follow the argument here. If both of your operators in eqn (3) are 'min', then why in Algorithm (1) is an argmax performed? If one wanted to minimise the sum of minimal distances (min-of-min, as eqn (3) shows), why not take argmin?
> > >
> > > Let's consider (2) and (3) separately.
> > >
> > > (2) is simply finding the farthest point (with respect to the sampled nodes). Thus, we need the argmax function to find the point with the maximum distance.
> > >
> > > The cost function of (2) measures how accurate a set of sampled points cover a set of data points. The cost value for each data point is equal to its distance to the closets (min) sampled data points. Now, we minimized (min) this cost function to find the best set of sampled points.
> > >
> > > 12. “Okay, this clarifies my doubt. Could you please make it more clear somewhere in the paper that you're using exactly Equation (1) in your implementation?”
> > >
> > > We did. Thanks.

---

> > > > ### Comment · AnonReviewer4 · 2019-11-14
> > > > **Score upgrade**
> > > >
> > > > Thank you for clarifying all the issues -- now everything is clear.
> > > >
> > > > I have just upgraded my score to 6. Good luck!

---

> > > > > ### Author Response · Authors · 2019-11-15
> > > > > **Thank you.**
> > > > >
> > > > > We sincerely thank the reviewer for the suggestions and his/her positive feedback.

---

### Decision · Program_Chairs · 2019-12-19

**Decision:**

Reject

**Comment:**

The authors identify a limitation of aggregating GNNs, which is that global structure can be mostly lost. They propose a method which combines a graph embedding with the spatial convolution GNN and show that the resulting GNN can better distinguish between similar local structures.

The reviewers were mixed in their scores. The proposed approach is clearly motivated and justified and may be relelvant for some graphnet researchers, but the approach is only applicable in some circumstances - in other cases it may be desirable to ignore global structure. This, plus the high computational complexity of the proposed approach, mean that the significance is weaker. Overall the reviewers felt that the contribution was not significant enough and that the results were not statistically convincing.  Decision is to reject.

---

> ### Author Response · Authors · 2020-01-27
> **Response**
>
> We thank the chair for handling our paper. These concerns were addressed in details to the reviewers and that is why they increased their scores after we responded to their comments.
>
> - "but the approach is only applicable in some circumstances - in other cases it may be desirable to ignore global structure."
> Our approach helps the GNN to distinguish the local structures and also prevents the GNN from mapping different local structures from each other. In addition, graph classification requires the GNN to understand both local and global structures of the given graph. The requirements of graph classification are completely different from those of node classification. As we showed through several numerical experiments, mapping the local structures to similar feature vectors and not understanding the global structure of the graph, can make the GNN completely unable to perform even simple classification tasks.
>
> - "high computational complexity of the proposed approach": Our approach does not have high computation complexity. The complexity of the embedding step is linear with the number of nodes.
>
>
>
> - "not significant enough": Table 1 and Table 2 show up to 100 % improvement in the results.